# MV140 Mucosal Vaccine Induces Targeted Immune Response for Enhanced Clearance of Uropathogenic *E. coli* in Experimental Urinary Tract Infection

**DOI:** 10.3390/vaccines12050535

**Published:** 2024-05-14

**Authors:** Paula Saz-Leal, Marianne Morris Ligon, Carmen María Diez-Rivero, Diego García-Ayuso, Soumitra Mohanty, Marcos Viñuela, Irene Real-Arévalo, Laura Conejero, Annelie Brauner, José Luis Subiza, Indira Uppugunduri Mysorekar

**Affiliations:** 1Inmunotek S.L., 28805 Madrid, Spain; cmdiez@inmunotek.com (C.M.D.-R.); dgarcia@inmunotek.com (D.G.-A.); mvinuela@inmunotek.com (M.V.); ireal@inmunotek.com (I.R.-A.); lconejero@inmunotek.com (L.C.); jlsubiza@inmunotek.com (J.L.S.); 2Department of Obstetrics and Gynecology, Washington University School of Medicine, St. Louis, MO 63110, USA; marianne.ligon@gmail.com (M.M.L.); indira.mysorekar@bcm.edu (I.U.M.); 3Department of Microbiology, Tumor and Cell Biology, Karolinska Institutet, 17177 Stockholm, Sweden; sm14@nibmg.ac.in (S.M.); annelie.brauner@ki.se (A.B.); 4Division of Clinical Microbiology, Karolinska University Hospital, 17176 Stockholm, Sweden; 5Department of Medicine, Section of Infectious Diseases, Baylor College of Medicine, Houston, TX 77030, USA; 6Department of Molecular Virology and Microbiology, Baylor College of Medicine, Houston, TX 77030, USA

**Keywords:** MV140, UTI89, mucosal immunity, vaccine, UTI—urinary tract infection, UPEC—uropathogenic *E. coli*

## Abstract

MV140 is an inactivated whole-cell bacterial mucosal vaccine with proven clinical efficacy against recurrent urinary tract infections (UTIs). These infections are primarily caused by uropathogenic *E. coli* (UPEC) strains, which are unique in their virulence factors and remarkably diverse. MV140 contains a non-UPEC strain, suggesting that it may induce an immune response against different UPEC-induced UTIs in patients. To verify this, we experimentally evaluated the cellular and humoral responses to UTI89, a prototypical UPEC strain, in mice vaccinated with MV140, as well as the degree of protection achieved in a UPEC UTI89 model of acute cystitis. The results show that both cellular (Th1/Th17) and antibody (IgG/IgA) responses to UTI89 were induced in MV140-immunized mice. MV140 vaccination resulted in an early increased clearance of UTI89 viable bacteria in the bladder and urine following transurethral infection. This was accompanied by a highly significant increase in CD4^+^ T cells in the bladder and an increase in urinary neutrophils. Collectively, our results support that MV140 induces cross-reactive humoral and cellular immune responses and cross-protection against UPEC strains.

## 1. Introduction

Urinary tract infections (UTIs) are one of the most common bacterial infections, infecting an estimated 150 million people worldwide annually [1]. This makes them a major health concern, which is underscored by their frequency and the potential severity of their symptoms, varying from painful urination and abdominal pain to critical kidney infections if not treated [1,2]. While these infections are commonly treated with antibiotics, growing antibiotic resistance is complicating treatment as some UTIs become less responsive to conventional therapies [3].

UTIs mostly affect women, significantly impacting their quality of life. More than half of all women will have at least one UTI in their lifetime, and despite treatment with antibiotics, 20–30% of these women will suffer a recurrence within three to four months of the initial infection [1,2]. Recurrent UTIs may require long-term antibiotic prophylaxis, which has numerous side effects and contributes to the emergence of resistant strains, which in turn favors further recurrences [4,5]. The frequent use of antibiotics, coupled with antibiotic resistance among uropathogens, highlights the urgent need to develop new and improved treatment and prevention options [6]. Although vaccination to prevent the occurrence of UTIs is an attractive approach, the development of effective vaccines against UTIs has proven to be more complex than expected [7,8].

Uropathogenic *Escherichia coli* (UPEC) is the most common etiologic agent of UTIs, accounting for approximately 80% of cases [9]. UPEC strains possess unique characteristics and virulence factors that distinguish them from other pathogenic or non-pathogenic *E. coli* strains [10,11,12]. On the other hand, UPEC strains show remarkable diversity due to their strain-specific adaptations to different habitats [13], which is favored by their higher mutation rates compared with other *E. coli* strains [14]. This diversity could pose a challenge to the development of anti-UPEC vaccines [15], especially when targeting single antigens; as such, vaccines may not be effective against the full spectrum of UPEC strains [16]. In this regard, the use of whole-cell bacterial vaccines, rather than subunit vaccines, containing both conserved and strain-specific proteins, is expected to provide the most comprehensive protection in UTIs [17].

In UTIs, the primary entry point for uropathogens is the urothelium, which is part of the bladder mucosa. This mucosa constitutes the initial barrier resisting infection [18]; thus, enhancing this barrier through vaccination could be a strategic goal. For this purpose, mucosal vaccines are preferable over parenteral vaccines, as they elicit stronger mucosal immunity even at distant mucosal sites [19]. In particular, the sublingual and intranasal routes are effective in stimulating effector responses in the urogenital mucosa due to the interconnected nature of the mucosal-associated lymphoid tissues [20].

MV140 is a whole-cell bacterial mucosal (sublingual) vaccine containing *E. coli* and other species of bacteria associated with UTIs [21]. Despite the fact that the *E. coli* strain (V121) in MV140 was originally derived from a non-pathogenic strain, MV140 prevents recurrent UTIs in various clinical settings [22,23,24,25,26,27,28,29,30,31,32]. A randomized, placebo-controlled trial demonstrated its efficacy in preventing uncomplicated recurrent UTIs in women [33]. Thus, clinical data suggest that MV140 provides cross-protection against UPEC strains that cause UTIs in patients. Although this supports the existence of a cross-reactive immune response between MV140 and different UPEC strains, there is no experimental evidence to confirm this in models of protection against UTIs. Preclinical studies showed that MV140 induces antibody and CD4^+^ T cell responses against MV140-contained bacteria following sublingual or intranasal vaccination [34,35].

The purpose of this study was to evaluate the immune response induced by MV140 against UTI89, a prototypical UPEC strain not found in MV140. Additionally, this study aimed to evaluate the protection conferred by MV140 after a subsequent challenge with UTI89 in an experimental model of UTI. Our results indicate that mucosal immunization with MV140 induces local and systemic antibody and T-cell responses against UTI89 and enhances the clearance of UTI89 from the bladder and urine after experimental infection.

## 2. Materials and Methods

### 2.1. Mice and In Vivo Models

Female C57BL/6 Wild-type mice (8- to 10-week-old) from Jackson laboratory were used. Mice were maintained under specified pathogen-free conditions in a barrier facility under a 12-h light–dark cycle. All experimental procedures were approved by the animal studies committee of Washington University in St. Louis School of Medicine (Animal Welfare Assurance #A-3381-01).

As immunization protocol, mice were intranasally (i.n.) administered with 50 μL of MV140, a suspension of equal amounts of whole-cell heat-inactivated bacteria: V121 *Escherichia coli*, V113 *Klebsiella pneumoniae*, V125 *Enterococcus faecalis*, V127 *Proteus vulgaris* (Uromune^®^, Inmunotek S.L., Madrid, Spain) at 300 Formazin Turbidity Units (FTUs, ~10^9^ bacteria/mL), or vaccine excipients (control) diluted in phosphate-buffered saline (PBS; Sigma-Aldrich, St. Louis, MO, USA), 3 times a week for 2 weeks, and rested for an additional week. One week after the last immunization, some mice were sacrificed to collect serum samples and spleens for ex vivo stimulation assays. Other mice were kept alive for urine collection over the next 7 days to quantify IgA antibodies.

For the experimental UTI, UTI89, a clinical UPEC isolate from a patient with recurrent cystitis [36] was grown statically for 17 h in Luria–Bertani broth (LB, Tryptone 10 g/L, Yeast extract. 5 g/L. and NaCl 10 g/L, Sigma-Aldrich) at 37 °C prior to infection. One week after the last immunization, mice were anesthetized and inoculated via transurethral catheterization with 10^7^ colony-forming units (CFUs) of UTI89 in PBS, as published elsewhere [37]. Urines and bladders were collected at indicated timepoints (24–72 h post-infection, h.p.i) and spotted onto LB-agar plates to measure bacterial titers.

### 2.2. Splenocyte Ex Vivo Stimulation Assay

T cell responses were assessed in the spleen of MV140-immunized or control mice. Spleens were processed to obtain single-cell suspensions following conventional protocols. Cells (1.5 × 10^6^/mL) were plated in 96-well plates (200 μL final volume; Corning, Corning, NY, USA) and in vitro stimulated for 72 h with heat-killed UTI89, MV140 (both 3 FTU, ~10^7^ bacteria/mL) or control, and cytokine production (IL-17A, IFN-γ, IL-4, and IL-10) was measured in cell-free culture supernatants by Multiplex cytokine assay (Bio-plex, Bio-Rad Laboratories, Ann Arbor, MI, USA), following the manufacturer’s instructions.

### 2.3. Serum and Urine Antibody Measurement

IgG and IgA antibody levels for MV140, MV140-containing *E. coli* strain (V121)*,* or UTI89 were determined by Enzyme-Linked Immunosorbent Assay (ELISA) from serum and urine samples obtained at indicated timepoints. Briefly, 96-well non-tissue culture-treated plates (Greiner Bio-One, Monroe, NC, USA) were pretreated with poly-L-lysine (Sigma-Aldrich) for 1 h under UV light. Then, plates were coated with the heat-inactivated MV140, V121 *E. coli* or UTI89 (all at 450 FTU) overnight at 4 °C, and, subsequently, incubated with 50 µL of mouse serum for 2 h at room temperature (RT). In the case of the urine samples, 50 µL of pooled concentrated urine were used per well. After a washing step, Igs were detected using a horseradish peroxidase-labeled antibody (anti-Mouse IgG or IgA from Sigma). After a final washing step, the peroxidase substrate (OPD, 3 mg/mL from Sigma-Aldrich) was added in 0.1 M citrate buffer with 0.03% H_2_0_2_—pH 5.5. The enzymatic reaction was allowed to develop for 30 min and stopped by adding a 10% HCl solution. Plates were read at 492 nm (Synergy Mx, Biotek, Agilent Technologies, Durham, NC, USA), and antibody concentration was expressed as arbitrary units (AUs) per mL, calculated from the optical density (OD) at 492 nm using the following formula: AU/mL = [(OD × dilution factor] × 10.

### 2.4. Flow Cytometry

Bladders were digested at 37 °C for 30 min in RPMI-1640 with 10 mM HEPES, collagenase D, and DNAse (all from Sigma-Aldrich), forced through a 70 μm cell strainer (Corning), and washed with 5% fetal bovine serum (FBS, ThermoFisher Scientific, Durham, NC, USA) in PBS. Urines were diluted in fluorescence-activated single-cell sorting (FACS) Buffer (PBS supplemented with 5 mM EDTA and 3% FBS) for 30 min (pH neutralization), prior to staining. Single-cell suspensions resuspended on ice-cold FACS Buffer were stained with anti-CD45 eFluor450 (eBioscience, San Diego, CA, USA), anti-CD3-A700 (eBioscience), anti-CD4-PE/Cy7 (BioLegend, San Diego, CA, USA), anti-CD8-BrilliantViolet605 (BioLegend), anti-CD11b-PE/Cy5 (eBioscience), anti-Ly6C-APC/Cy7 (BioLegend), anti-Ly6G-FITC (BioLegend), anti-F4/80-PE (eBioscience), anti-CD64-APC (BioLegend), and 7-AAD (BioLegend). Purified anti-FcɣRIII/II (BioLegend) was used to block Fc-receptors. Data were acquired on LSR II flow cytometer (BD) and analyzed with FlowJo software v10.0 (BD).

### 2.5. Antimicrobial Peptide Production by RT-qPCR and Immunofluorescence

For RT-qPCR, bladders were flash-frozen and RNA was extracted using TRIzol reagent (Invitrogen, San Diego, CA, USA) according to the manufacturer’s protocol, followed by gDNA digestion with TURBO DNA-free kit (Invitrogen). cDNA was generated using Superscript II Reverse Transcriptase (Invitrogen). qPCR was performed with standard SYBR green (Applied Biosystems, San Diego, CA, USA) on a Rotor-Gene PCR cycle (Corbett Life Science, Mortlake, Australia). Gene-specific primers Cramp (forward: AATTTTCTTGAACCGAAAGGGC, reverse: TGTTTTCTCTCAGATCCTTGGGAGC), S100a7a (forward: GCTCGTTTAGTGAACCGTCAG, reverse: GGAGTCCTCCACTGGTGTGT), and Actb (forward: CTGTCCCTGTATGCCTCTG, reverse: ATGTCACGCACGATTTCC) were used. Fold changes were calculated using Ct method and normalized internally to the respective control.

For immunofluorescence analysis, bladders were fixed in methacarn (60% methanol, 30% chloroform, 10% acetic acid) and embedded in paraffin. Bladder sections were deparaffinized using neoclear and rehydrated, pretreated with 0.3% Triton X-100 in PBS (PBS-T), and boiled in citrate buffer, 1 mM EDTA, 10 mM TRIS, 0.05% Tween 20 (pH 9). Sections were rinsed with PBS-T, treated with FX Signal Enhancer (Invitrogen) at RT for 30 min and blocked for 60 min with sera. Furthermore, anti–psoriasin (1:200; Santacruz) and anti-UPIIIa (1:200; Santa Cruz) were incubated overnight at 4 °C. Sections were washed with PBS-T and further incubated with respective secondary Alexa Fluor-conjugated antibodies (Invitrogen) in 1:600 for 1.5 h at RT, followed by staining with DAPI for 15 min, washed and mounted in Fluoromount G (Southern Biotech, Birmingham, AL, USA). Slides were analyzed with a Zeiss 700 confocal microscope and fluorescence intensity quantified with the ImageJ Fiji 1.53b software.

### 2.6. Statistical Analysis

The statistical analysis was performed using Prism (GraphPad Software v.8.0.2.). Statistical significance for comparison between treatment groups was determined using Mann–Whitney or unpaired Student’s *t* tests, according to normal distribution evaluated by Shapiro–Wilk test, or mixed-effects models. Outliers were identified by means of Tukey’s range test. For antibody analysis, arbitrary units were calculated from the optical density (OD) values according to standard curves, after subtracting the OD values of blank and adjusting by the serum dilutions performed. Differences were considered significant at *p* < 0.05. Except when specified, only significant differences are shown.

## 3. Results

### 3.1. MV140 Induces T and B Cell Immune Responses Cross-Reactive with UTI89

Mice were immunized intranasally with MV140 and the immune response to both MV140 and UTI89 was assessed as shown in Figure 1A. The T-cell response was evaluated in vitro using splenocytes stimulated with either bacteria by analyzing the production of the corresponding cytokines, Th1 (IFN-γ), Th2 (IL-4), Th17 (IL-17), and regulatory T-cell (IL-10). As shown in Figure 1B, MV140-immunized mice produced significantly higher levels of IFN-γ, IL-17, and IL-10 than control mice after stimulation with MV140 or UTI89. Notably, comparable levels of these cytokines were measured in MV140-immunized mice regardless of the bacteria (MV140 or UTI89) used as stimuli. Thus, the response to IL-17 was strongly induced by both stimuli, whereas IL-4 was not induced by either stimulus (Figure 1B). Likewise, mice immunized with MV140 produced IgG antibodies reacting with MV140, V121 (i.e., the *E. coli* strain in MV140), and UTI89 (Figure 1C). Moreover, a significant IgA response reacting with UTI89 was also observed in the serum and urine of MV140-immunized mice compared to control mice (Figure 1C).

### 3.2. MV140 Confers Protection against Transurethral UTI89 Infection

Transurethral infection with UTI89 is a well-established murine model of acute urinary infection [37]. To test whether MV140 conferred protection in this model, viable UTI89 bacteria were administered to mice immunized with MV140 or the excipient as a control (Figure 2A). The course of infection was assessed by determining the bacterial load during the acute phase (1–3 days post-infection). As shown in Figure 2B, the bacterial load was significantly lower in both the urine and bladder of MV140-immunized mice compared to control mice one day after infection. Consistent with the expected outcomes in this infection model [36,37], the bacterial load was greatly reduced in both immunized and control mice by day 3 post-infection.

### 3.3. MV140 Induces Changes in the Local Inflammatory Cell Influx after Transurethral UTI89 Infection

The local inflammatory response in the urine and bladder one day after infection with UTI89 was compared between mice immunized with MV140 and controls. As shown in Figure 3A, there was a significant increase in CD45^+^ cells, mainly neutrophils and monocytes, in the urine of MV140-immunized mice at 24 h post-infection. At this time point, there was also a trend toward increased CD4^+^ T cells in the urine. This increase, with respect to the CD45^+^ compartment, was highly significant in the bladder of MV140-immunized mice compared to non-immunized mice (Figure 3B).

The possible differential induction of antimicrobial peptides by the urothelium 24 h after infection was also evaluated. Specifically, the expression of CRAMP (cathelin-related antimicrobial peptide) and psoriasin (S100a7a), which have been shown to be relevant as resistance mechanisms in UTIs [38,39], were analyzed by qPCR. As shown in Figure 3C, there was an increase bordering on statistical significance (*p* = 0.051) in psoriasin expression in MV140-immunized mice compared to control mice, but not in CRAMP. The increase in psoriasin expression could not be detected at the protein level by immunofluorescence analysis of the urothelium at this time point (Appendix A).

### 3.4. UTI89 Infection Boosts IgG Responses against Different E. coli Strains in MV140-Immunized Mice

Finally, we assessed the humoral response two weeks after infection to evaluate whether a challenge infection with UTI89 boosts the humoral response induced by the MV140 vaccine. As expected, serum IgG levels against all MV140 bacteria, as well as against the *E. coli* strain V121 comprised in MV140, and against UTI89, were higher in mice immunized with MV140 on day 1 post-infection (Figure 4). As can be seen, antibody levels against V121 and UTI89 both demonstrated a noticeable increase 14 days after infection in mice immunized with MV140. This suggests that a UTI89 infection may enhance the response of *E. coli*-specific antibodies in mice that have been previously vaccinated.

## 4. Discussion

UPEC belongs to different *E. coli* phylogroups, each of which is characterized by numerous virulence factors and genes that enhance their pathogenicity and resistance to antimicrobial treatment with the emergence of a multidrug-resistant phenotype [40]. This situation underscores the critical need for novel treatment strategies and preventive actions, including vaccination.

MV140 is a mucosal vaccine that has been shown to prevent UTI recurrences in different clinical settings, including a randomized placebo-controlled clinical trial [21,22,23,24,25,26,27,28,29,30,31,32,33]. Given the diversity of UPEC strains responsible for the majority of UTIs, MV140 is expected to exhibit broad-spectrum activity against them. The purpose of this study was to address this possibility experimentally. To this end, the immune response to a prototypic UPEC strain (UTI89) in mice immunized with MV140 and its effect in conferring protection against infection with the same UPEC strain were assessed. MV140 contains V121, clonally derived from a non-pathogenic *E. coli* strain ascribed to phylogenetic group B1, while UTI89 is a UPEC strain from a cystitis isolate ascribed to group B2 [13,36].

Transurethral infection with UTI89 is a well-established acute UTI model in mice [13,36,37]. Here, we show that mucosal (intranasal) immunization with MV140 results in an early and significant reduction in bacterial load in the urine and bladder compared to non-immunized controls. This was accompanied by an increased local influx of myeloid and CD4^+^ T cells in MV140-immunized mice. The rapid increase in T cells in the bladder following UTI89 challenge suggests a recall response due to prior vaccination [41], thus supporting cross-reactivity between MV140 and UTI89. While T-cell responses are considered serotype-independent [42], cross-reactive T-cell epitopes are frequent even between different enterobacteria species causing UTIs [43].

A similar cytokine response (Th1/Th17) was obtained when splenocytes from MV140-immunized mice were stimulated with MV140 or UTI89. These T cell responses were supported by previous preclinical studies with MV140, both in vitro and in vivo [34,35,44]. T cells, particularly Th17, are thought to play a key role in the control of mucosal infections caused by extracellular bacteria [45]. Activated Th17 cells control the influx of neutrophils into mucosal tissues by releasing inflammatory cytokines such as TNF-α and IL-17, which in addition may induce the production of neutrophil-attracting chemokines by epithelial cells [46] and antimicrobial peptides [47]. In a previous study, the production of TNF-α, but not IL-17, was detected ex vivo in isolated bladder cells from MV140-immunized mice in response to MV140 [34]. In the present study, psoriasin, an antimicrobial peptide synergistically induced by IL-17/IFN-γ [48] and involved in host resistance to UTIs [39], was detected as increased at the mRNA level (*p* = 0.051) in the bladder of only MV140-immunized mice after their challenge with UTI89.

Other T cell responses (IL-10 or IL-4 production) obtained in MV140-immunized mice also showed a similar pattern upon stimulation with MV140 or UTI89. While the IL-10 response was significantly increased in MV140-immunized mice, as was the case for IFN-γ or IL-17, no response (and even a decreasing trend) was observed for IL-4. The lack or down-regulation of Th2-derived cytokines (IL-4/IL-5) was previously noted when analyzing Th2 cell-polarizing properties of human monocyte-derived DCs primed with MV140 [34,35]. This effect may be relevant in the context of UTI, given the detrimental role attributed to Th2 responses sustained by recurrent infections in bladder cell resistance to *E. coli* [49]. In this context, the Th1 and IL-10 responses induced by MV140 immunization may contribute to the suppression of pre-existing Th2 responses [45].

In mice immunized with MV140, serum IgG antibody levels against V121 or UTI89 were comparable. Following a challenge with UTI89, there was a noticeable increase in antibody responses against both bacteria. This suggests the existence of a common antigenic repertoire for B cells after immunization with MV140. Although V121 and UT189 show different serotypes (e.g., O6 vs. O18, respectively) shared B cell epitopes are likely when considering whole-cell bacteria. It is therefore possible that the antibody response may also contribute to the cross-protective mechanisms induced by MV140. Moreover, MV140 vaccination-induced IgA antibodies reactive with UTI89 were detected in serum but also in the urine, pointing to their local production. Interestingly, local IgA is favored by tissue-resident lymphocytes derived from Th17 cells induced upon vaccination [50].

In this study, we demonstrate that MV140 confers protection against a prototypical UPEC strain in an experimental model of UTI by inducing a cross-reactive immune response. However, it is not clear whether other mucosal bacterial vaccines against recurrent UTIs containing *E. coli* that have reached the clinic will induce the same type of response, as their clinical performance varies in terms of efficacy and duration [51]. Factors such as the composition of the bacterial mixture, formulation, and/or route of administration influence the response elicited [52]. Finally, it is important to note that, although this study is a proof of concept with an experimental model of acute UTI, the T-cell response generated by MV140 could help to breach the described vicious circle associated with recurrent UTIs [21,49].

## 5. Conclusions

This study demonstrates that mucosal immunization with MV140 induces robust T/B cell responses and provides cross-protection against a prototypic UPEC strain (UTI89) not included in MV140. These findings support the potential of MV140 as a broadly effective vaccine against a wide range of UPEC strains responsible for urinary tract infections (UTIs). Moreover, the specific immune responses induced by MV140, including Th1/Th17 and IL-10 but excluding IL-4, suggest its potential as an important tool in preventing and managing recurrent UTIs.

## Figures and Tables

**Figure 1 vaccines-12-00535-f001:**
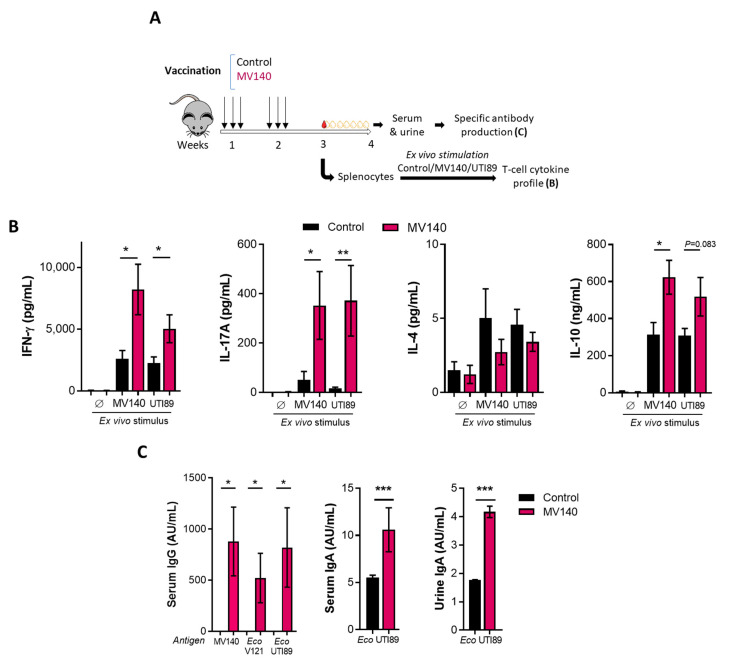
Mucosal vaccination with MV140 induces the generation of local and systemic adaptive immune responses. (**A**) Scheme of the intranasal immunization protocol with vaccine excipients, (control, black) or MV140 (magenta) and analysis of induced immune response. (**B**) Cytokine production (IFN-γ, IL-4, IL-17A, and IL-10) in supernatants of splenocytes isolated from mice immunized and restimulated according to A. Mean ± SEM of two independent experiments (n = 7) is shown. (**C**) Serum/urine-specific IgG (left) and IgA antibodies against MV140, MV140-containing *E. coli (Eco)* V121, or uropathogenic *E. coli (Eco)* UTI89 generated in mice according to A. Data are shown as arbitrary units per mL (AU/mL), calculated as described in the Methods Section. Mean ± SEM of 3 replicates from pooled samples shown (n = 8). (**B**,**C**) * *p* < 0.05, ** *p* < 0.01, *** *p* < 0.001, unpaired Student’s t-test or Mann–Whitney test comparing between treatment groups, according to normal distribution assessed using Shapiro–Wilk test. Ø, unstimulated.

**Figure 2 vaccines-12-00535-f002:**
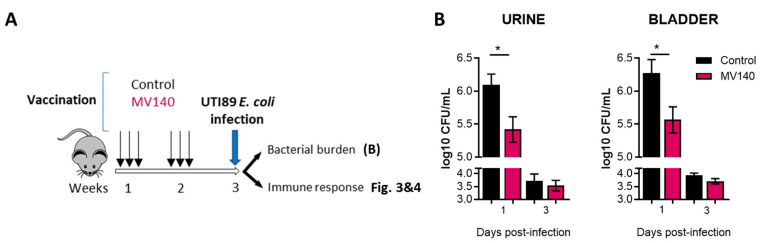
MV140 mucosal vaccine confers early protection against uropathogenic *E. coli* UTI89 infection. (**A**) Scheme of the intranasal immunization protocol and subsequent transurethral infection with uropathogenic *E. coli* UTI89. (**B**) Bacterial load in the urine (left panel) and bladder (right panel) at indicated timepoints upon infection. Mean ± SEM of three (bladder) or eight (urine) independent experiments is shown (n ≥ 8). * *p* < 0.05, Mann–Whitney test, according to the absence of normal distribution assessed by Shapiro–Wilk test. p.i., post-infection. CFUs, colony-forming units.

**Figure 3 vaccines-12-00535-f003:**
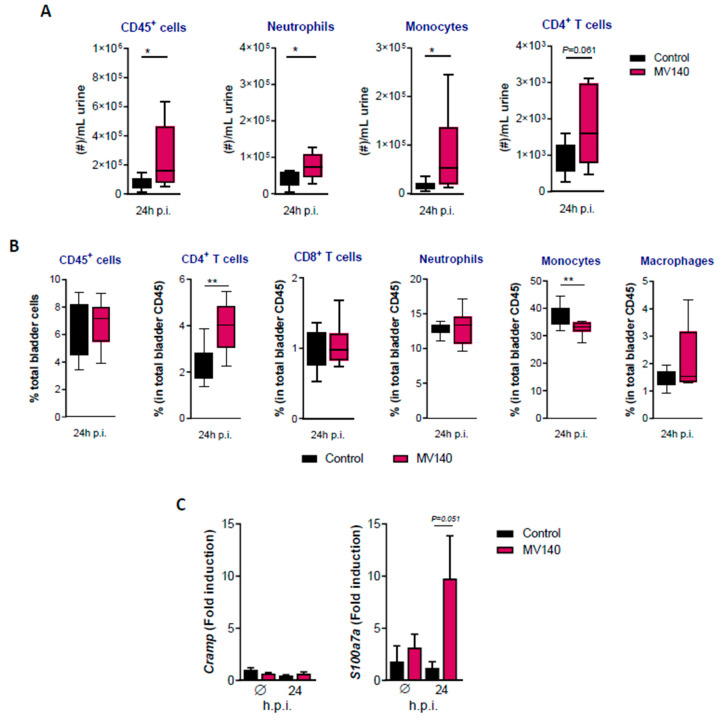
MV140 mucosal vaccine induces local cellular immunity following uropathogenic *E. coli* UTI89 infection. (**A**,**B**) Cellular response including CD45^+^ cells, neutrophils, monocytes, macrophages, and T lymphocytes in urines (**A**) and bladders (**B**) at 24 h post-infection, analyzed by flow cytometry. Data are shown as boxplots and min–max whiskers of two independent experiments (n ≥ 8). (**C**) Antimicrobial peptide relative expression (*CRAMP*, *S100A7A*) in total bladder at 24 h post-infection, analyzed by q-PCR. Fold induction vs. control uninfected mice is represented. Mean + SEM of one (uninfected) or two (24 h post-infection) independent experiments is shown (n ≥ 2). (**A**–**C**) Mice were immunized with control (vaccines excipients, black) or MV140 (magenta) and subsequently infected as stated in Figure 2A. * *p* < 0.05, ** *p* < 0.01, unpaired Student’s *t*-test or Mann–Whitney test, according to normal distribution assessed by Shapiro–Wilk test. Ø, uninfected; p.i., post-infection.

**Figure 4 vaccines-12-00535-f004:**
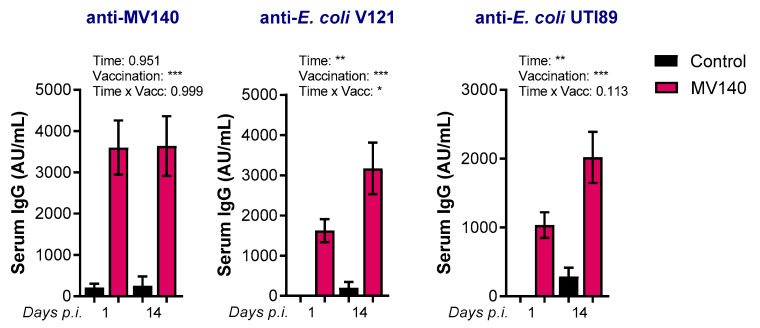
MV140 mucosal vaccine confers heightened humoral response following uropathogenic *E. coli* UTI89 infection. Specific serum IgG antibodies against MV140, MV140-containing *E. coli* V121, and uropathogenic *E. coli* UTI89 at indicated timepoints post-infection. Data are shown as arbitrary units per mL (AU/mL), calculated as described in the Methods Section. Mean ± SEM of 2–3 independent experiments per timepoint (n ≥ 10). Mice were immunized with control (vaccines excipients, black) or MV140 (magenta) and subsequently infected as stated in Figure 2A. * *p* < 0.05, ** *p* < 0.01, *** *p* < 0.001, mixed-effects model. Time and vaccination are variables considered independently as single factors (time or vaccination) or simultaneously (time x vacc) for the statistical analysis.

## Data Availability

The data can be shared up on request.

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
