# Peer review of "MV140 Mucosal Vaccine Induces Targeted Immune Response for Enhanced Clearance of Uropathogenic E. coli in Experimental Urinary Tract Infection"

_vaccines, 2024, doi:10.3390/vaccines12050535_

Round 1
Reviewer 1 Report
Comments and Suggestions for Authors
Congratulations to the authors, I enjoyed reading their manuscript. I think the experimental design is perfect for the intended purpose and the results of this study highlight what they observed in women vaccinated with MV140.
I believe that mouse models are a useful and cost-effective tool for vaccine validation studies and vaccine-induced immune response studies.
I think the manuscript is well written and the graphs are very intuitive and illustrative of the results. The discussion is well done and enjoyable, and the conclusion is concise and gives a clear answer to the stated objective.
I have nothing to suggest. I repeat my congratulations.
Author Response
We really appreciate the reviewer for taking time to carefully review the manuscript and give such positive comments. These constructive words keep us motivated in our research and our way of doing science. Thank you very much again.
Reviewer 2 Report
Comments and Suggestions for Authors
MV140 is a whole-cell bacterial mucosal vaccine containing E. coli and other species of bacteria associated with urinary tract infections (UTIs).
The manuscript by Paula Saz-Leal et al reported an interesting study that evaluated experimentally the immune response induced by MV140 against UTI89, a prototypical UPEC strain not comprised in MV140. Their results indicated that mucosal immunization with MV140 induces local and systemic antibody and T cell responses against UTI89 and enhances clearance of UTI89 from the bladder and urine after experimental infection.
Overall, the study was well designed, the data were clearly presented, and the manuscript was very well written.
I have no significant criticisms.
A minor error:
Line 259: UT89 - should be UTI89.
Author Response
We are very grateful to the reviewer for the positive comments given on the experimental design and the manuscript. As suggested, we modified the typo in the revised version of the manuscript.
Thank you very much again.
Reviewer 3 Report
Comments and Suggestions for Authors
I have read with interest the manuscript submitted by Saz-Leat et al, since AMR represents a global concern and any method of preventing infections is of great interest.
The manuscript is well-written and organized;
My only remarks would be:
- all Latin bacterial names should be italicized;
- figure 2 has low resolution;
- can the authors mention if there were any side effects encountered? This should be included for further transparency, especially since there is mentioned a conflict of interest.
- the discussion and conclusion sections could be expanded;
- the reference list is relevant and edited according to the mdpi pattern. Avoid self-citation in such a high number, include only the necessary articles.
Best regards.
Author Response
We really appreciate the time and effort dedicated to providing such a valuable feedback on the manuscript.
Here is a point-by-point response to the specific comments and concerns.
- All Latin bacterial names should be italicized.
All latin names have been revised and italicized in the revised version of the manuscript. Thank you for the comment.
- Figure 2 has low resolution
As rightly pointed out by the reviewer, we noticed that Figure 2, particularly panel 2A might have low resolution. A renewed version of the figure is provided in this revised version of the manuscript. Thank you again to the reviewer for the comment.
- Can the authors mention if there were any side effects encountered? This should be included for further transparency, especially since there is mentioned a conflict of interest.
Regarding the reviewer concern, no side effects were encountered in any of the mice following immunization or any other procedure. Of note, this immunization setup is well established for both MV140 and other bacterial vaccines in experimental models of infection [1,2].
Furthermore, the safety of MV140 has been widely demonstrated in humans from a randomized controlled trial and over 10 years of real-world evidence in the clinical setting [3-5].
- The discussion and conclusion sections could be expanded;
Following the reviewer’s suggestion, discussion (Lines 279-283, page 8; Lines 335-334, page 9) and conclusions (Lines 346-352, page 9) have been expanded. As is the introduction (Lines 36-52, pages 1-2; Lines 64-70, page 2), to address the important topic of AMR and mucosal immunity, and some methodological details (Lines 134-135, page 3; Lines 160-170, page 4) we considered of interest for the reader. Thank you again for the recommendation.
- The reference list is relevant and edited according to the mdpi pattern. Avoid self-citation in such a high number, include only the necessary articles.
We sincerely appreciate the reviewer's comments. We have used 52 references in the revised manuscript. Of these, 7 are self-citations that we feel are important to support the information provided in the text. The remaining citations related to MV140 (9 in total) are from third party authors who have published mainly clinical data supporting the relevance of MV140 in various settings. We sincerely believe that all of them are relevant to contextualize the previous data and provide the reader with insights into the results of MV140 in both preclinical and clinical studies.
REFERENCES
- Del Fresno, C.; Garcia-Arriaza, J.; Martinez-Cano, S.; Heras-Murillo, I.; Jarit-Cabanillas, A.; Amores-Iniesta, J.; Brandi, P.; Dunphy, G.; Suay-Corredera, C.; Pricolo, M.R.; et al. The Bacterial Mucosal Immunotherapy MV130 Protects Against SARS-CoV-2 Infection and Improves COVID-19 Vaccines Immunogenicity. Front Immunol 2021, 12, 748103, doi:10.3389/fimmu.2021.748103.
- Brandi, P.; Conejero, L.; Cueto, F.J.; Martinez-Cano, S.; Dunphy, G.; Gomez, M.J.; Relano, C.; Saz-Leal, P.; Enamorado, M.; Quintas, A.; et al. Trained immunity induction by the inactivated mucosal vaccine MV130 protects against experimental viral respiratory infections. Cell Rep 2022, 38, 110184, doi:10.1016/j.celrep.2021.110184.
- Lorenzo-Gómez, M.-F.; Foley, S.; Nickel, J.C.; García-Cenador, M.-B.; Padilla-Fernández, B.-Y.; González-Casado, I.; Martínez-Huélamo, M.; Yang, B.; Blick, C.; Ferreira, F.; et al. Sublingual MV140 for Prevention of Recurrent Urinary Tract Infections. NEJM Evidence 2022, 1, EVIDoa2100018, doi:doi:10.1056/EVIDoa2100018.
- Ramirez Sevilla, C.; Gomez Lanza, E.; Llopis Manzanera, J.; Cetina Herrando, A.; Puyol Pallas, J.M. A Focus on Long-Term Follow-Up of Immunoprophylaxis to Recurrent Urinary Tract Infections: 10 Years of Experience with MV140 Vaccine in a Cohort of 1003 Patients Support High Efficacy and Safety. Arch Esp Urol 2022, 75, 753-757, doi:10.56434/j.arch.esp.urol.20227509.110.
- Nickel, J.C., Saz-Leal, P., Doiron, R.C. Could sublingual vaccination be a viable option for the prevention of recurrent urinary tract infection in Canada? A systematic review of the current literature and plans for the future. CUAJ 2020, 14, 281-287.